# Molecular Dynamics Simulation of Polyacrylamide Adsorption on Cellulose Nanocrystals

**DOI:** 10.3390/nano10071256

**Published:** 2020-06-28

**Authors:** Darya Gurina, Oleg Surov, Marina Voronova, Anatoly Zakharov

**Affiliations:** G.A. Krestov Institute of Solution Chemistry of the Russian Academy of Sciences, 1 Akademicheskaya St., 153045 Ivanovo, Russia; miv@isc-ras.ru (M.V.); agz@isc-ras.ru (A.Z.)

**Keywords:** cellulose nanocrystals, polyacrylamide, adsorption, molecular dynamics

## Abstract

Classical molecular dynamics simulations of polyacrylamide (PAM) adsorption on cellulose nanocrystals (CNC) in a vacuum and a water environment are carried out to interpret the mechanism of the polymer interactions with CNC. The structural behavior of PAM is studied in terms of the radius of gyration, atom–atom radial distribution functions, and number of hydrogen bonds. The structural and dynamical characteristics of the polymer adsorption are investigated. It is established that in water the polymer macromolecules are mainly adsorbed in the form of a coil onto the CNC facets. It is found out that water and PAM sorption on CNC is a competitive process, and water weakens the interaction between the polymer and CNC.

## 1. Introduction

Polymer nanocomposites are materials that consist of polymer matrices and nanofillers distributed in them. The most important factor for achieving the enhancement of the bulk physical (transport, thermal, mechanical) properties of nanocomposites is interfacial interactions between the polymer macromolecules and the nanofiller particles. A large number of nanofillers with a variety of polymer matrices have been used to achieve the enhancement of useful properties. The simplified model of a nanofiller reinforced polymer suggests the wrapping of the polymer macromolecules around the nanofiller, providing an intuitional understanding of the mechanism responsible for the improved physical properties. However, large-scale disorder in polymer nanocomposites leads to substantial reduction of physical properties compared to predictions based on idealized filler morphology [1,2]. Computer simulations can address the issues related to the difficulty of characterizing the in situ structure of nanoscale objects [3,4]. Achievements in modeling polymer matrix nanocomposites based on identification of some challenges can enhance our understanding of physicochemical properties of polymer nanocomposites [5]. Specifically, in-depth understanding of structure and dynamics at solid/polymer interfaces at nanometer level plays a key role in designing materials with adjustable properties [6,7,8].

The features of cellulose nanocrystals (CNC) (a high crystallinity degree, anisotropic shape, high aspect ratio, and large surface area) have important consequences for the interfacial behavior and have attracted great attention of the materials community because of their sustainability, renewability, and biodegradability. Therefore, CNC have widely been used as reinforcing agents in polymeric nanocomposites in recent years [9].

Nowadays, atomistic modeling of cellulose nanocrystals has been used to complement experimental measurements. Computer simulations help to predict self-assembly as well as mechanical, energetic, thermal, and structural features of cellulosic nanomaterials and provide a fundamental understanding of the atomic-scale origins of these characteristics [10,11,12,13,14,15,16]. Models have been employed to predict some CNC properties including the most frequently reported mechanical ones [17,18]. Molecular modeling has been utilized to investigate the properties of amorphous cellulose as opposed to CNC [19,20,21]. Some studies have applied molecular modeling to investigate the interaction of CNC with a liquid solvent (most solvent studies are focused on water) [22,23,24] and with polymeric materials [25].

The current work is a continuation of our previous study that was devoted to the investigation of the microscopic mechanism of polyvinylpyrrolidone (PVP) adsorption on a cellulose nanocrystal and the role of water in this process [26]. We have revealed earlier [27] that PVP adsorption onto CNC can block lateral interactions between the CNC and prevents their agglomeration in the lateral direction, i.e., hinders growth of the CNC particles width upon the concentration increase or drying of the composites. Moreover, freezing of CNC suspensions can align rod-like CNC particles in direction of ice crystal growth allowing formation of the CNC aggregates with a high aspect ratio. These aggregates can be broken down easily in water and some organic solvents, providing good dispersibility of the composites.

In this work we focus our attention on the intermolecular interactions in the systems containing polyacrylamide (PAM), CNC and a polar solvent (water). PAM is a water-soluble linear polymer. PAM and its derivatives have different applications in many areas including agricultural, water treatment, medical, and petroleum industries [28,29]. PAM differs from PVP because it contains not only atoms playing the role of proton-acceptors, but also atoms acting as proton-donors. Because of such structure the behavior of PAM in water, as well as its interactions with the solvent and CNC have some features that we describe in terms of structural and dynamic properties by the all atomic molecular dynamics method.

## 2. Computational Details

Classical molecular dynamics simulations were carried out using a GPU-accelerated (graphics processing unit) Gromacs-5.0.7 software package [30]. Molecular graphics and visualization were performed using VMD 1.8.6 [31]. The molecular dynamics (MD) simulations were carried out for the NVT ensemble (a constant number of particles N, a volume V, and a temperature T). The reference temperature of 298 K was kept constant using a Nose–Hoover thermostat [32,33] with the coupling constant τ = 0.1 ps. Periodic boundary conditions were applied to all three directions of the simulated cubic box. The Verlet algorithm [34] was adopted to integrate the equations of motion. The modified Ewald summation method [35,36] was used to account for the corrections of the long-range electrostatic interactions with a cutoff radius of 1.5 nm, which was also the cutoff value for the Van der Waals (VDW) interactions. All the bond length constraints were implemented using the LINCS algorithm (LINear Constraint Solver) [37]. In our work for PAM (Figure 1a) we utilized a potential based on OPLSAA (Optimized Potential for Liquid Simulations All Atomic) force field parameters [38]. The initial structure of a PAM macromolecule containing 50 monomer units (with the molecular weight of 3552 g/mol) was constructed by means of Avogadro [39]. This number of monomer units is sufficient to observe the conformational transitions of the polymer and to obtain sufficient statistical data to calculate quantitative characteristics. For cellulose we used GROMOS54a7 force field parameters [40]. The initial structure of CNC was built based on the experimental crystallographic data [41] by a toolkit named Cellulose Builder [42]. The model of the I_β_ CNC consisted of 14 or 9 glucan chains, and the degree of polymerization of each chain was 10 (Figure 1b,c). The number of chains and degree of polymerization provide a sufficient surface area for effective interactions with the polymer. As known, in the structure of cellulose there is clear segregation into polar (OH) and nonpolar (CH) sites [10]. Because of the hydrophobic properties of the glucopyranose plane, the sheet-like structure of the top and bottom surfaces (200) of CNC (Figure 1c–e) has a predominantly hydrophobic character. CNC with predominantly hydrophilic surfaces (110 and 1–10) (Figure 1h, for example) has a large number of free OH groups (O1H1, O6H4, O3H3, O2H2). In the current study, we consider two types of CNC with large hydrophobic facet (200) and with large hydrophilic facets (110 and 1–10). The systems were solvated by water which was preliminarily equilibrated in *NpT*-ensemble at 298 K and 0.1 MPa. For water molecules, the SPC/E (extended simple point charge) model [43] was used. After systems energy minimization, we equilibrated the systems for 0.5 ns in the *NVT* ensemble. The production run simulations were performed in *NVT* ensemble in a cubic box with periodic boundary conditions for 10 ns for all systems in a vacuum and binary system PAM-water, 15 ns for large systems with a high concentration of PAM. For the systems with a low PAM concentration, the simulation time was chosen as 60 ns to obtain sufficient statistical data. A time step was set of 1 fs. The data for the analysis were collected every 0.1 ps. In Supplementary Material one can find the time dependencies of temperature for all systems, which are evidence that the structures obtained are equilibrated well at all length- and time-scales (Appendix A).

Ten systems were simulated; the simulation details are listed in Table 1. The initial configuration of Systems 2 was constructed placing 1 PAM molecule with a nearly extended conformation into a cubic box with 6916 water molecules. Initial Systems 3, 5, 7, and 9 were built by inserting one CNC with predominantly hydrophobic and hydrophilic surfaces, respectively, into the center of the cubic boxes. Then 3 or 64 PAM macromolecules were placed into the cell with CNC, which was followed by the addition of 31322 (System 3 and System 5) or 273740 (System 7 and System 9) water molecules. The size of the water box was chosen so that it ensured the systems had at least a 20 Å solvation shell in all the directions. As an example the instant snapshot of System 3 is presented in Figure 1i. The initial configurations of Systems 4, 6, 8, and 10 were obtained by removing the water molecules from the final configuration of Systems 3, 5, 7, 9, respectively.

## 3. Results and Discussion

In order to understand the structural behavior of the polymer chain placed into the water box, we calculated the radius of gyration (*R*g), which is one of the most important quantities in conformational statistics of polymer chains [44,45]. The radius of gyration was calculated by the following equation:(1)$$R_{g}={\left(\frac{{\sum_{i}\left\|r_{i}\right\|}^{2}m_{i}}{\sum_{i}m_{i}}\right)}^{1/2},$$
where *m*_i_ is the mass of site *i* and *r*_i_ is the position of site *i* relative to the center of mass of the molecule. As can be seen in Figure 2, *R*_g_ differs from the original value in both the vacuum (System 1) and water (System 2). In water, the value of the radius of gyration fluctuates during the simulation process and achieves ≈ 1.5 nm, while at the beginning of the simulation *R*_g_ was 1.7 nm. Therefore, we can conclude that we observe a certain transformation of the polymer chain, which, however, does not lead to a strong folding of the PAM and the chain remains quite expanded, like a coil. In the absence of a polar solvent (water), the polymer chain folds quickly into a dense conformation (polymer globule), and *R*_g_ becomes equal to approximately 1.0 nm. The snapshots of the last simulation frame, presenting the PAM coil in water (System 2) and the PAM globule in a vacuum (System 1), are also depicted in Figure 2. It should be noted that in the systems with a high PAM concentration (Systems 7 and 9), the average value *R*_g_ slightly decreases compared with the dilute solution containing only 1 PAM macromolecule and equals approximately 1.2 nm. This result is in agreement with the works of Chen and coauthors [46,47], where the value *R*_g_ = 1.20 nm was obtained for a water solution of a PAM macromolecule with a polymerization degree of 50 at 298 K.

As it may be supposed, changes in the PAM conformation in water or in a vacuum is connected with the PAM ability to form intra- and intermolecular hydrogen bonds (HBs). Using the geometrical criterion of HBs (a donor–acceptor distance less than 0.35 nm, and an acceptor–donor–hydrogen angle less than 30°), we calculated the time dependence of PAM–PAM and PAM–water HBs per one monomer unit of PAM in Systems 1 and 2 (Figure 3). Because of PAM folding in the vacuum, the number of HBs increases and at the end of the simulation almost every monomer unit forms one HB with another one. In contrast, in the presence of water, the number of intramolecular HBs between the PAM monomers is significantly smaller and is about 0.3. Although, as we have shown (Figure 2), in water the PAM chain exists in the form of a coil, a certain transformation is observed. The number of intramolecular HBs increases (Figure 3a), and the number of intermolecular PAM–water HBs decreases (Figure 3b). Nevertheless, by the end of the tenth nanosecond, each monomer unit of the polymer forms an average of 2 HBs with the water molecules. Table 2 lists some structural and dynamic characteristics, including average numbers of HBs, for all the systems. 

The stability of a HB can be characterized by HB lifetimes τ_HB_. The continuous mean lifetime of a HB [48], *τ*_HB_, was calculated from the autocorrelation function (ACF) *C*_HB_(*t*) of the parameter characterizing the HB existence between the *i* and *j* molecules *S_ij_*(*t*) using the standard program package:(2)$$C_{HB}^{}\left(t\right)=\frac{\langle S_{ij}(0)S_{ij}(t)\rangle }{\langle S_{ij}^{2}(0)\rangle },$$
where *S_ij_*(*t*) = 1 if the criterion of HB existence between the *i* and *j* molecules was satisfied at the initial moment of time and is satisfied at a current moment of time *t*, and if the duration of the violated criterion periods on the time interval from 0 to *t* did not exceed a predetermined value *t**; otherwise, *S_ij_*(*t*) = 0. At *t** = 0, the autocorrelation function *C*_HB_(*t*) demonstrates continuous existence of hydrogen bonds. Integration of this function using the standard program package gives the mean HB lifetimes:(3)$$\tau_{HB}^{}=\int_{0}^{\infty }C_{HB}^{}(t)dt,$$

Table 2 shows that the PAM–water HB lifetime is longer than the PAM–PAM τ_HB_. Moreover, in water the lifetime of PAM–PAM HBs is shorter than in a vacuum.

In order to understand which sites of molecules take part in HB formation, we calculated the atom–atom radial distribution functions (RDFs). Figure 4 shows the radial distribution functions of oxygen Ow and hydrogen Hw water atoms around the polymer atoms in System 2. Figure 4a compares five intermolecular RDFs for pairs of PAM atoms with water molecules, including H6-Ow, H7-Ow, O4-Hw, N5-Ow, and O4-Ow, where H6, H7, N5, and O4 represent hydrogen, nitrogen, and oxygen atoms of the PAM, respectively (Figure 1a). In Figure 4a, two peaks appear in the vicinity of the carbonyl oxygen, O4, for g_O4-Hw_(r) and g_O4-Ow_(r). These peaks are positioned at approximately 0.19, 0.33 nm and 0.29, 0.54 nm for g_O4-Hw_(r) and g_O4-Ow_(r), respectively, and represent two hydration shell structures forming around O4. The first peak at 0.19 nm in the g_O4-Hw_(r) indicates a high probability of the HB formation between O4 and water molecules. At the same time the pre-peaks at 0.24 nm and 0.25 nm in the g_H6-Ow_(r) and g_H7-Ow_(r) (which also satisfy the HB geometric criterion R_H---O_ ≤ 0.26 nm) confirm hydrogen bond formation. However, the height and position of the peaks allow us to suppose the existence of weaker H6-Ow and H7-Ow HBs compared with O4-Hw. Besides, the height of the first peak of g_O4-Ow_(r) is greater than that of g_N5-Ow_(r), which also means that hydration of oxygen is stronger than that of nitrogen. The behavior of the C1-Ow, C2-Ow, H89-Ow, and H11-Ow RDFs, namely, the existence of small shoulder peaks at long distances, can be attributed to a highly hydrophobic nature of the PAM hydrocarbon chain. Thus, a comparison of the RDFs confirms that the HBs are predominantly formed through the PAM carbonyl oxygen and only a few of them are formed through the amide group hydrogen atoms. In Singh’s work [49], the DFT computations were performed on various acrylamide-water clusters in the gas phase in order to explore the microsolvation. In clusters consisting of one acrylamide molecule and 5–15 water molecules, the length of the Ow---H-N HB was ≈ 0.21 and the length of the Hw---O=C HB varied between 0.18 and 0.19 nm. This observation is in agreement with our molecular dynamics results. Also the authors of [49] calculated the binding energy of the acrylamide–acrylamide and acrylamide–water hydrogen-bonded complexes and obtained the values of 6.52 and 12.48 kcal/mol per an acrylamide molecule, respectively. This result is consistent with our findings for the HB lifetime of PAM–PAM and PAM–Water interactions, and explains why PAM macromolecules tend to form intermolecular HBs with water molecules rather than intramolecular HBs.

To prove the existence or absence of the interaction between PAM and CNC in Systems 3–10, we calculated the time evolution of the number of contacts (Figure 5) between any pair of PAM and CNC atoms within a given distance (0.5 nm). In Systems 3, 7, and 9, the adsorption process was successful, and all the three PAM macromolecules in System 3, four PAM molecules in System 7, and one PAM molecule in System 9 were adsorbed on the CNC. In System 3 after 5 ns of the modeling, the number of close contacts increased and at the end of the simulation reached ≈200 (Figure 5a and Table 2). At the same time in System 5, where the CNC surface possesses a predominantly hydrophilic character with a large number of OH-groups, only a few rare episodes of interactions between PAM and CNC were observed (Figure 5a and Table 2). On the contrary, in a vacuum (Figure 5b,d) the PAM molecules were adsorbed on both the hydrophobic (Systems 4, 8) and hydrophilic facets (Systems 6 and 10) of CNC. The number of close contacts in a vacuum was significantly bigger than that in water. This implies that there were competitive interactions in Systems 3, 5, 7, and 9, since the molecules of water and PAM were able to form hydrogen bonds with the oxygen and hydrogen atoms of CNC. Because of the high hydrophilicity of CNC in Systems 5 and 9, almost all the free OH-groups of cellulose were involved in HB formation with the water molecules. Table 2 shows that an average number of HBs for CNC–water was bigger and their lifetime was longer in Systems 5 and 9 than in Systems 3 and 7. Moreover, the CNC–water hydrogen bonds were slightly more stable than those between the water molecules. In general, if we analyze the hydrogen bond lifetimes in all the systems, then we can arrange HBs in the order of increasing stability as follows: *τ*_HB_(PAM–PAM) < *τ*_HB_(PAM–Water) < *τ*_HB_(PAM–CNC) < *τ*_HB_(Water–Water) < *τ*_HB_(CNC–Water) < *τ*_HB_(CNC–CNC). Cellulose chains had a strong affinity with each other and with materials containing hydroxyl groups. Therefore, the lifetimes of the CNC–CNC and CNC–water HBs were higher than the lifetimes of other HBs in the systems. In the work of Chen et al. [50], it was found that water-cellulose HBs are on average stronger than water–water ones. In particular, they found that E_HB_(Water–Water) ~ 4.2 kcal/mol, while E_HB_(Water–cellulose) ~ 5.4 kcal/mol. Having compared τ_HB_(PAM–CNC) for Systems 3 and 4, we can suppose that water weakens the interaction of the polymer with the cellulose nanoparticles.

In order to visualize the process of the polymer adsorption on cellulose, we presented snapshots of the modeling cells for Systems 3–10 at the end of the simulation (Figure 6). To highlight CNC and PAM, all the water molecules from Systems 3, 5, 7, and 9 were removed. In System 3, two PAM macromolecules were adsorbed on the top hydrophobic side of CNC and one macromolecule was adsorbed on the bottom hydrophobic side of CNC. The same picture was observed for System 4, i.e., for PAM–CNC composite in a vacuum. The losses of CNC–water intermolecular hydrogen bonds were compensated with new hydrogen bonds with PAM, and we observed a significant increase in <n_HB_> for PAM–CNC (Table 2). On the other hand, although PAM molecules interacted with a hydrophilic facet of CNC more actively in a vacuum than in water, a large part of the CNC surface remained free, and the number of PAM–CNC close contacts in System 6 was three times less than in System 4. Besides, the average minimal distance of the PAM contact with a CNC hydrophilic facet, <*r*_min_>, was approximately 1.3 times longer than with a hydrophobic one (Table 2). At the same time, in the systems with a high polymer concentration (Systems 7–10), the difference between the average numbers of closest contacts was not so noticeable for hydrophobic and hydrophilic CNC, i.e., the probability to find PAM near CNC significantly increased and almost all the accessible HB centers of CNC were occupied by PAM. Table 2 clearly shows that the average numbers of PAM–CNC HBs per one CNC monomer in Systems 8 and 10 were equal to 0.76 and 1.03, respectively, which is ≈2.4 and ≈4.7 times higher than in the systems with a deficiency of PAM molecules. The difference between the <N_C_> values for Systems 8 and 10 (1519 and 1387, respectively) is due to the fact that the area of the available surface for the polymer is smaller for hydrophilic CNC. Additionally, with gmx_sasa tool from Gromacs-5.0.7 we calculated the total accessible solvent surface area (SASA) for both CNC types: SASA (hydrophobic CNC) = 73.1 nm^2^ and SASA (hydrophilic CNC) = 58.4 nm^2^.

Figure 7 shows the places of the polymer direct contact with the cellulose in the systems. It can be seen that in the presence of the solvent the contact areas between the polymer and cellulose are much smaller than in the vacuum. Most of the CNC solvation shell in System 3 is water molecules (Figure 7).

It should be noted that in our previous work devoted to studying the adsorption of PVP on CNC, we also observed the crucial role of water in the interaction between PVP and CNC [26]. In particular, we showed that the presence of water makes the interaction between the polymer and cellulose weaker than in a vacuum, and the polymer and cellulose mainly interact through their solvation shells. Unlike PVP, which only has a hydrogen acceptor atom, PAM is able to act as a proton donor because of the amide group hydrogen atoms, which increases the probability of HB formation with the cellulose atoms. To characterize the interaction between PAM and CNC atoms in greater detail, we also analyzed RDFs for Systems 3–6 (Figure 8, Figure 9 and Figure 10). The RDFs for Systems 7–10 demonstrate the same behavior; that is why they are not shown here. As in the case of PAM–water interactions, the carbonyl oxygen O4 takes the most active part in the HB formation with the cellulose H1, H2, H3 atoms (Figure 8). The typical peaks, which can be found on the RDF curves g_H7O1_(r), g_H7O2_(r) at ≈0.23–0.24 nm, indicate that the H7 atom also actively interacts with CNC through the O1 and O2 atoms (Figure 8). 

In the absence of water (System 4) all the peaks on the RDF curves become more pronounced and increased compared with those for System 3, which confirms that more HBs are formed between the PAM and CNC in a vacuum. A dramatic increase in the peak heights on g_H7O2_(r), g_H7O3_(r), g_O4H2_(r), g_O4H3_(r), g_O4H4_(r) (Figure 9) is observed. It means that in water most of the cellulose hydroxyl groups O2H2 and O3H3 are involved in the HBs with the solvent molecules rather than the polymer ones. As for the amide hydrogen atoms, the probability of the H7 atom participation in HB formation is higher than for the H6 atom, which is located closer to the carbonyl oxygen O4 and may have a steric hindrance to the HB formation. System 6 does not have such preferences for HB formation through cellulose hydroxyl groups O2H2, O3H3, O1H1 (Figure 10), with the exception of the ended O6H4 hydroxyl group (Figure 1b).

## 4. Conclusions

Classical molecular dynamics simulations of polyacrylamide (PAM) adsorption on a cellulose nanocrystal (CNC) were carried out. Ten different systems containing PAM and CNC in a vacuum or in water were simulated. The average number of hydrogen bonds and lifetime of PAM–PAM, PAM–water, PAM–CNC, CNC–water, CNC–CNC, and water–water hydrogen bonds were calculated. The PAM radius of gyration, end-to-end distance, and the average number of close contacts and average minimum distance between any pair of atoms of PAM and CNC were determined. The atom–atom radial distribution functions for PAM, PAM–CNC in a vacuum and water were analyzed. It was confirmed that hydrogen bonds are predominantly formed through carbonyl oxygen of PAM and the –NH_2_ group is the less active participant of the HB-formation process in the systems. It was noted that PAM macromolecules tend to form intermolecular hydrogen bonds with water molecules rather than intramolecular ones. Moreover, even in the systems with a high PAM concentration the number of PAM–CNC hydrogen bonds is low, despite the fact that such HBs are more stable than PAM–PAM and PAM–water HBs. Based on the analysis of the PAM radius of gyration it can be concluded that the polymer spends more time in a more extended configuration in water and on the CNC surface (*R*_g_ ≥ 1.19 nm) than in a vacuum at some distance from the CNC (*R*_g_ ≤ 1.0 nm). The crucial role of water in the interaction between PAM and CNC was highlighted. It was found out that water and PAM sorption on CNC is a competitive process, and water weakens the interaction between the polymer and CNC.

## Figures and Tables

**Figure 1 nanomaterials-10-01256-f001:**
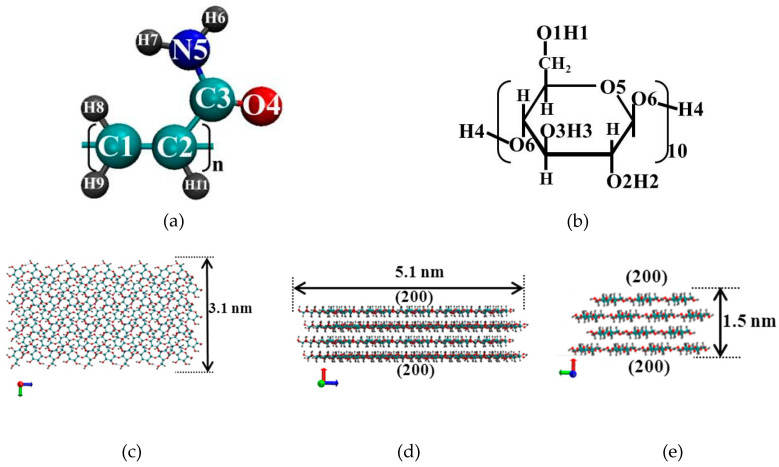
Atom numbering in monomer units of polyacrylamide (PAM) (**a**) and a cellulose nanocrystal (CNC) (**b**), CNC with a predominantly hydrophobic surface (**c**—**top view**; **d**,**e**—**end-on views**) and with a predominantly hydrophilic surface (**f**—**top view**; **g**,**h**—**end-on views**). The red spheres denote the oxygen atoms, the gray ones represent the hydrogen atoms, the cyan ones show the carbon atoms, and the blue sphere represents the nitrogen atom. The instant snapshot of System 3 (**i**): CNC is blue and PAM macromolecules are green, the red dots are the water molecules.

**Figure 2 nanomaterials-10-01256-f002:**
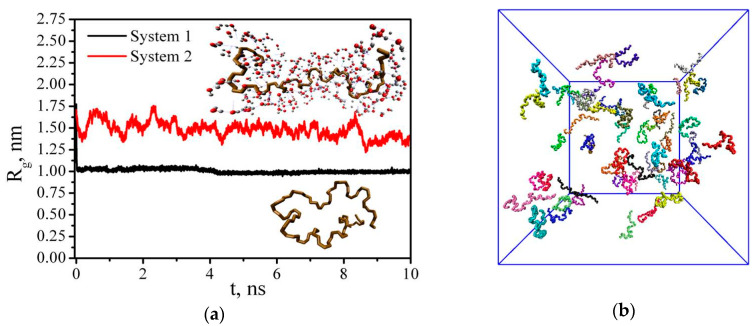
Time dependence of the radius of gyration of PAM (**a**) in a vacuum (System 1) and water (System 2), the snapshot of the PAM conformations in System 9 at the end of the simulation (**b**).

**Figure 3 nanomaterials-10-01256-f003:**
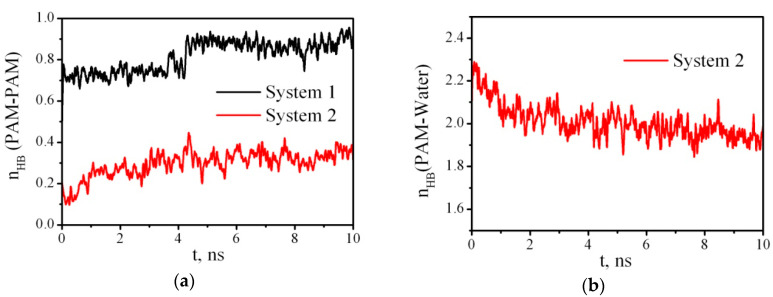
Examples of the time dependence of the number of PAM–PAM (**a**) and PAM–water (**b**) hydrogen bonds (n_HB_) per a PAM monomer unit.

**Figure 4 nanomaterials-10-01256-f004:**
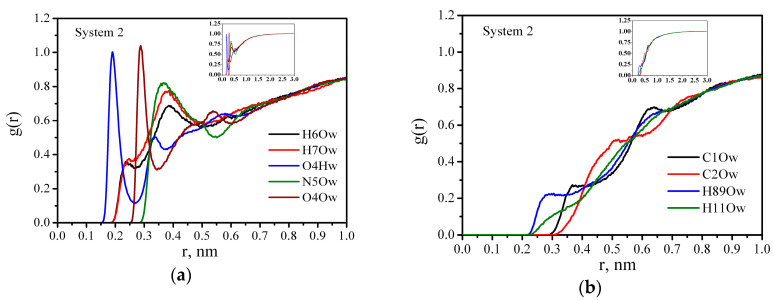
Atom–atom radial distribution functions for PAM in water: System 2. (**a**) The radial distribution functions of oxygen Ow and hydrogen Hw water atoms around the polymer atoms H6, H7, O4, N5, and O4 in System 2; (**b**) the radial distribution functions of oxygen Ow water atoms around the polymer atoms C1, C2, H8,9, and H11 in System 2.

**Figure 5 nanomaterials-10-01256-f005:**
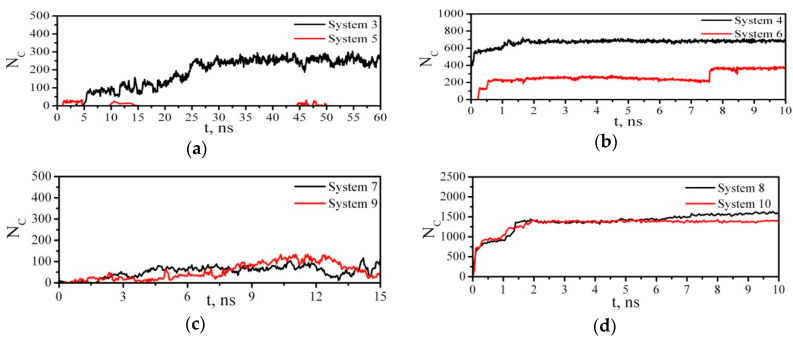
Time dependence of the number of contacts between any pair of atoms of PAM and CNC within a given distance of 0.5 nm. For System 5 when NC = 0, the red line coincides with the abscissa and therefore is not visible in the figure. (**a**) Systems 3 and 5, (**b**) Systems 4 and 6, (**c**) Systems 7 and 9, (**d**) Systems 8 and 10.

**Figure 6 nanomaterials-10-01256-f006:**
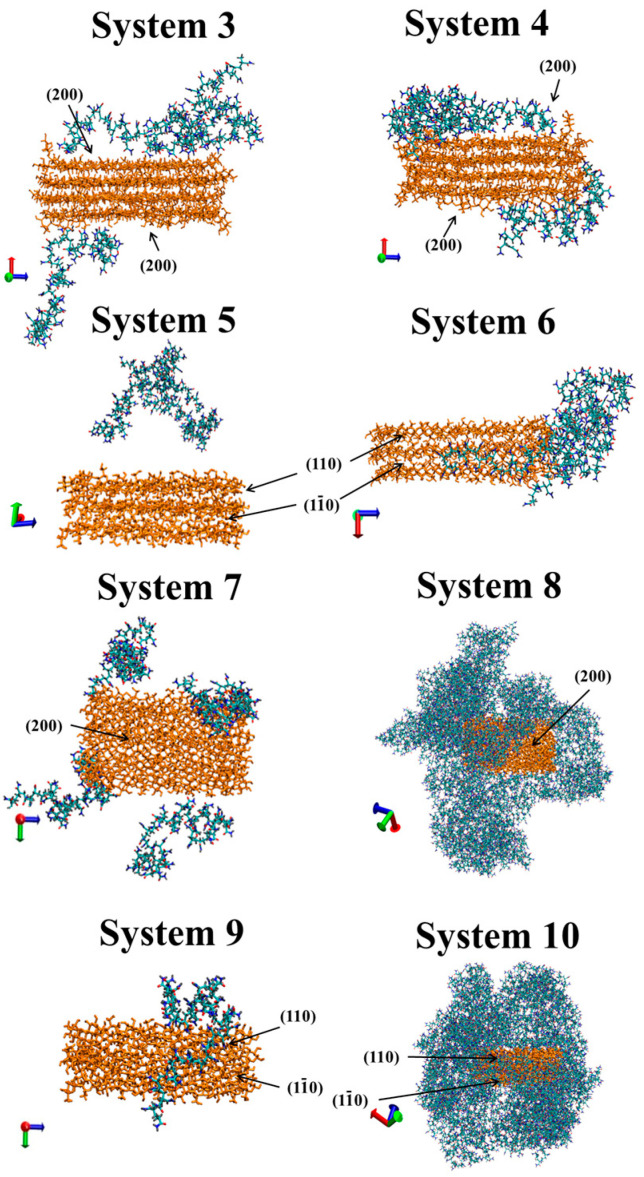
Snapshots of the last frame for Systems 3–10. The water molecules from Systems 3, 5, 7, and 9 were removed to highlight CNC (orange) and PAM. For better understanding, the hydrophobic (200) and hydrophilic (110, 1–10) facets are marked.

**Figure 7 nanomaterials-10-01256-f007:**
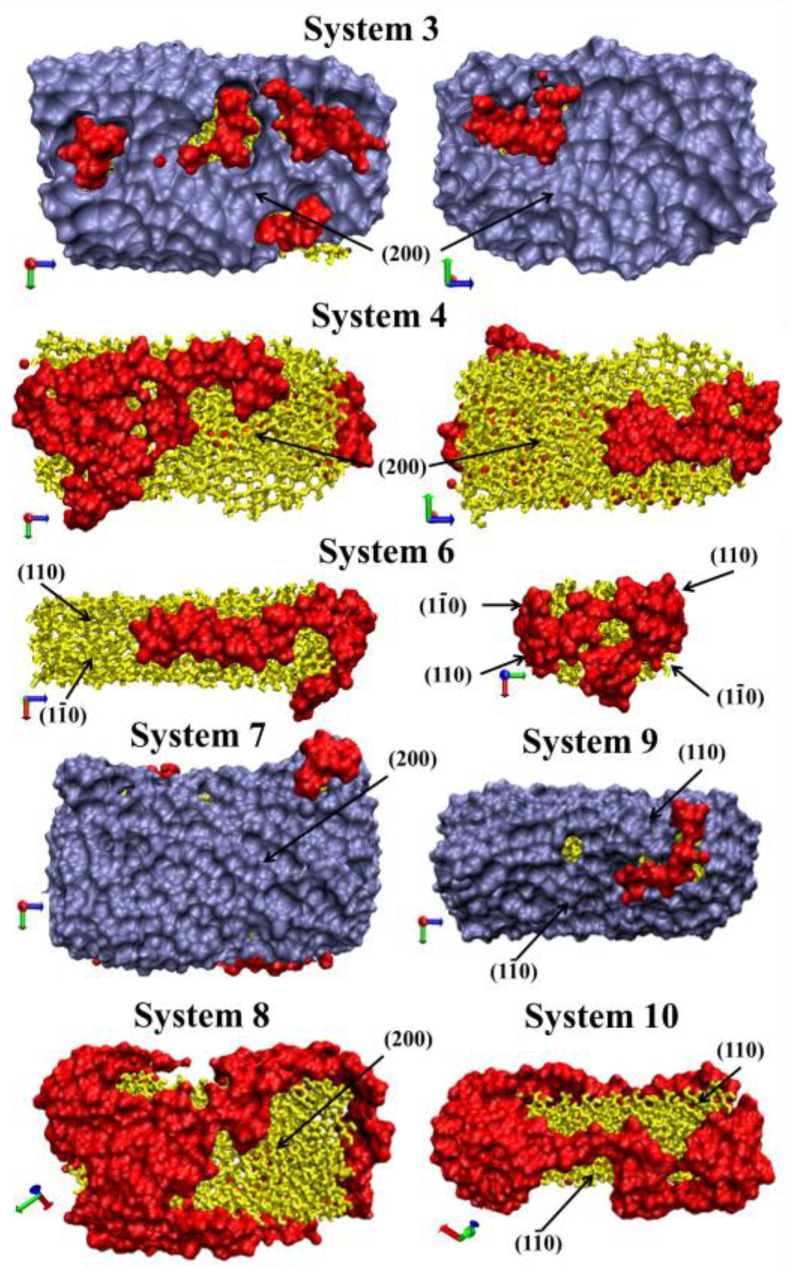
Snapshots of solvation shells at a distance of 0.5 nm from CNC (yellow) for System 3, System 4, System 6, System 7, System 9, System 8, System 10. The ice blue surface denotes the water shell and the red surface denotes the places of the PAM direct contact with CNC. For better understanding, the hydrophobic (200) and hydrophilic (110, 1–10) facets are marked.

**Figure 8 nanomaterials-10-01256-f008:**
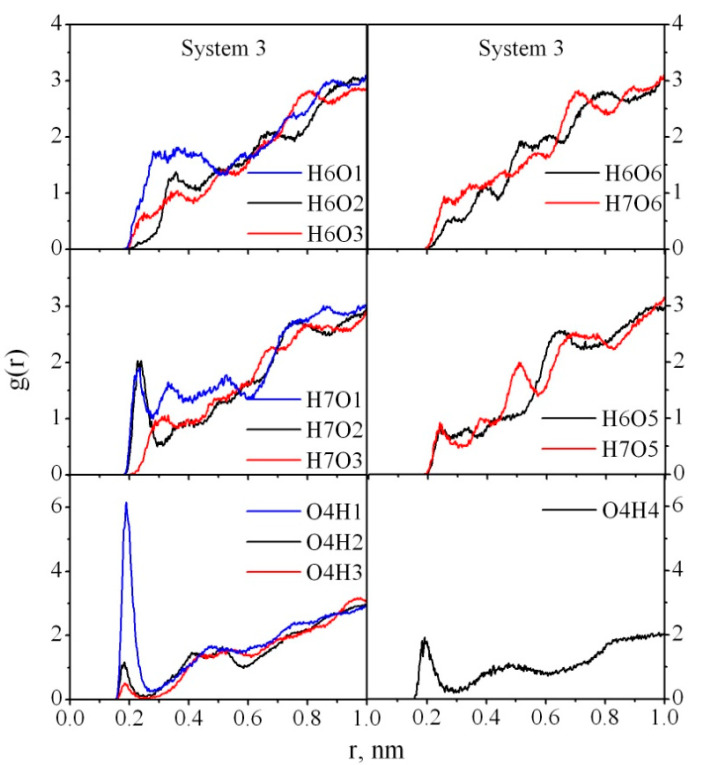
Atom–atom radial distribution functions for PAM–CNC in System 3.

**Figure 9 nanomaterials-10-01256-f009:**
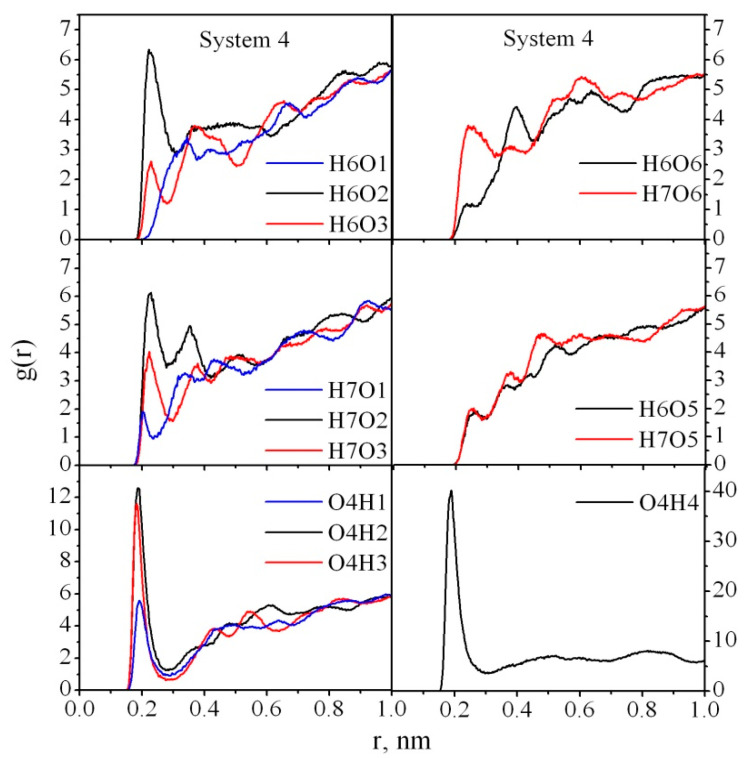
Atom–atom radial distribution functions for PAM–CNC in System 4.

**Figure 10 nanomaterials-10-01256-f010:**
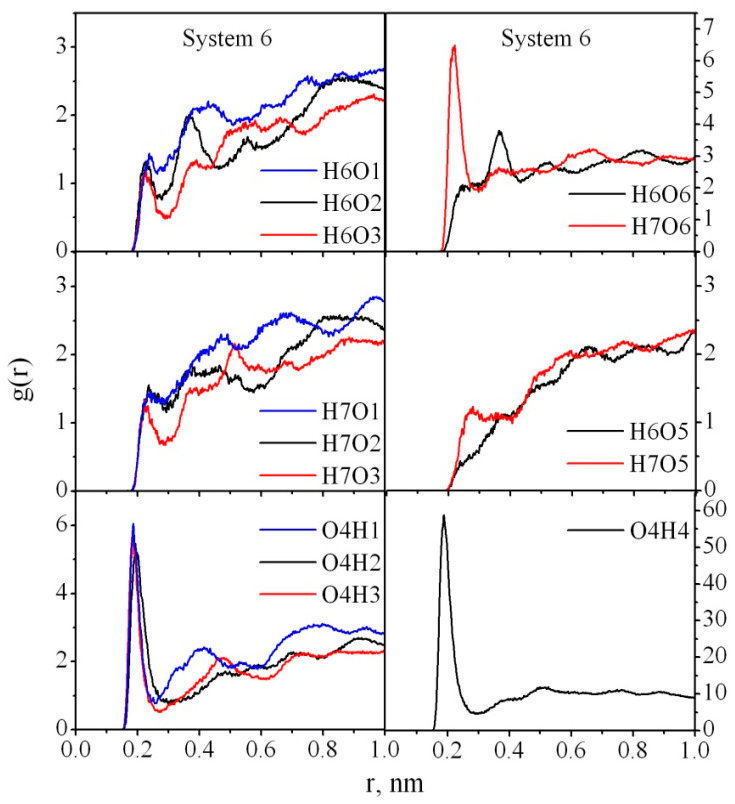
Atom–atom radial distribution functions for PAM–CNC in System 6.

**Table 1 nanomaterials-10-01256-t001:** Simulation details of different systems: the number (N) of PAM, CNC, and water molecules, simulation time (t, ns), length of box (L, nm).

System	Medium and CNC Type	N (PAM)	N (CNC)	N (Water)	t, ns	L, nm
System 1	vacuum	1	-	-	10	6.0
System 2	water	1	-	6916	10	6.0
System 3	water hydrophobic CNC	3	1	31322	60	9.0
System 4	vacuum hydrophobic CNC	3	1	-	10	9.0
System 5	water hydrophilic CNC	3	1	31322	60	9.0
System 6	vacuum hydrophilic CNC	3	1	-	10	9.0
System 7	water hydrophobic CNC	64	1	273740	15	20.5
System 8	vacuum hydrophobic CNC	64	1	-	10	20.5
System 9	water hydrophilic CNC	64	1	273740	15	20.5
System 10	vacuum hydrophilic CNC	64	1	-	10	20.5

**Table 2 nanomaterials-10-01256-t002:** The structural and dynamic characteristics obtained by analyzing the last 10 and 5 ns of the trajectories for water-contained and vacuum systems, respectively.

	System	1	2	3	4	5	6	7	8	9	10
<n_HB_>	PAM–PAM	0.76	0.30	0.35	0.66	0.40	0.76	0.31	1.00	0.32	1.03
	PAM–Water	-	2.01	1.80	-	1.83	-	1.93	-	1.94	-
	PAM–CNC	-	-	0.08	0.32	0.003	0.22	0.02	0.76	0.04	1.03
	CNC–Water	-	-	2.91	-	3.22	-	2.90	-	3.24	-
	CNC–CNC	-	-	3.61	4.26	3.31	3.88	3.66	4.3	3.40	3.86
	Water–Water	-	3.57	3.54	-	3.56	-	3.57	-	3.57	-
*τ, ps*	PAM–PAM	1.71	1.47	1.52	1.58	1.62	1.61	1.45	1.63	1.44	1.67
	PAM–Water	-	2.06	1.79	-	2.14	-	1.95	-	1.94	-
	PAM–CNC	-	-	2.40	2.85	-	2.83	2.37	3.07	2.14	2.81
	CNC–Water	-	-	3.19	-	3.87	-	3.21	-	3.86	-
	CNC–CNC	-	-	6.50	6.62	6.87	5.64	7.09	6.88	8.01	6.87
	Water–Water	-	2.81	2.77	-	2.77	-	2.77	-	2.77	-
*R*_g_, nm	PAM	1.00	1.48	1.28	1.19	1.34	1.30	1.21	0.95	1.22	0.96
*R*_e-t-e_*,* nm	PAM	0.99	3.56	3.55	3.04	3.56	3.75	2.78	1.75	2.79	1.81
*<N*c*>*	PAM–CNC	-	-	199	668	5	262	66	1519	73	1387
<*r*_min_> nm	PAM–CNC	-	-	2.07	1.67	2.19	2.17	2.06	1.62	1.97	1.63

<n_HB_> and τ are the average number of hydrogen bonds and lifetime of PAM–PAM and PAM–water hydrogen bonds (per a PAM monomer unit), PAM–CNC, CNC–water, and CNC–CNC ones (per a CNC monomer unit), and water–water ones (per a water molecule); *R*_g_ and *R*_e-t-e_ are the radius of gyration and the end-to-end distance of PAM, <*N*_C_> and <*r*_min_> are the average number of close contacts and average minimum distance between any pair of atoms of PAM and CNC within the limits of 0.5 nm.

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
