# Peer review of "Molecular Dynamics Simulation of Polyacrylamide Adsorption on Cellulose Nanocrystals"

_nanomaterials, 2020, doi:10.3390/nano10071256_

Round 1

Reviewer 1 Report

This manuscript detailed a set of simulations run to understand the interfacial behavior of a PAM-CNC nanocomposite. Overall, the manuscript is fairly well-written. However, I have the following suggestions, comments, and questions:

  1. It seems that this paper follows the template of a previous paper quite closely. Can any generalizations be made that negate the need to change the polymer matrix and publish a new set of simulations?
  2. The formatting of Table 1 can be improved by designating separate columns for the System name/ID, environment, and CNC type.
  3. In Figure 1, please state in the caption that the sub-figures are different views of the same molecule.
  4. Please explain the difference between the hydrophilic and hydrophobic CNC surface. How was this generated, and can the degree of each be quantified?
  5. Please clarify how the simulation time was determined. Why not have uniformity across the systems?
  6. In general, the Results and Discussion section could be more easily read if divided into sub-sections.
  7. In Figure 4, please use the same y-axis range and align sub-figures (a) and (b).
  8. In Figure 5(a), it is unclear if there is a red line (System 5) trace at 0, or if there is a reason for the trace to disappear completely.
  9. In Figure 5(b), what happens in System 6 at ~7.5 nm that results in a jump in N(C) after a period of stability?

Author Response

We are very grateful to the Reviewers for their work to review the manuscript, for the useful recommendations which helped to improve the manuscript.

Reviewer 1

This manuscript detailed a set of simulations run to understand the interfacial behavior of a PAM-CNC nanocomposite. Overall, the manuscript is fairly well-written. However, I have the following suggestions, comments, and questions:

  1. It seems that this paper follows the template of a previous paper quite closely. Can any generalizations be made that negate the need to change the polymer matrix and publish a new set of simulations?

Author response #1:

Our previous work and the investigations presented in the current work are parts of a project studying the nanocellulose composites with polymers of different nature. While PVP has only acceptor centers, PAM has a proton-donor amide group. Due to different nature of these polymers, it is logically to suppose that interactions existing in the systems CNC-polymer –water will have their own features. Firstly, a behavior of polymers in water will differ, and that may determine interactions with cellulose nanoparticles. That is why it is difficult to make any conclusions or generalizations without simulations.

  1. The formatting of Table 1 can be improved by designating separate columns for the System name/ID, environment, and CNC type.

Author response #2:

We thank the Reviewer for this recommendation. We tried to reformat Table 1, but it is getting too bulky considering an addition of another column with the length of the simulation cell.

  1. In Figure 1, please state in the caption that the sub-figures are different views of the same molecule.

Author response #3:

We agree with the comment. The figure caption was corrected as follows.

Figure 1. Atom numbering in monomer units of polyacrylamide (PAM) (a) and a cellulose nanocrystal (CNC) (b), CNC with a predominantly hydrophobic surface (c – top view, d, e – end-on views) and with a predominantly hydrophilic surface (f – top view, g, h – end-on views). The red spheres denote the oxygen atoms, the gray ones represent the hydrogen atoms, the cyan ones show the carbon atoms, and the blue sphere represents the nitrogen atom. The instant snapshot of System 3 (i): CNC is blue and PAM macromolecules are green.

  1. Please explain the difference between the hydrophilic and hydrophobic CNC surface. How was this generated, and can the degree of each be quantified?

 Author response #4:

As known, in the structure of cellulose there is a clear segregation into polar (OH) and nonpolar (CH) sites. Due to the hydrophobic properties of the glucopyranose plane, the sheet-like structure of the top and bottom surfaces of CNC (Figure 1c) has a predominantly hydrophobic character. CNC with a predominantly hydrophilic surface (Figure 1h, for example) has a large number of free OH groups (O1H1, O6H4, O3H3, O2H2). Both types of cellulose nanocrystals were built by Cellulose Builder toolkit based on the experimental crystallographic data. We have tried to estimate the surface area for both CNC types with gmx_sasa tool and obtained data are presented in the text (page 9, lines 255-256).

  1. Please clarify how the simulation time was determined. Why not have uniformity across the systems?

 Author response #5:

The simulation time was determined based on the system's size. Molecular dynamics simulations is time consuming and computationally expensive. The simulation time choosing by us for each system is the best balance between the computational cost and accuracy for our systems which allows us to obtain large statistical data for the following analysis. For example, in systems with low PAM concentration, the probability of PAM and CNC interaction is lower than that in the systems with a high concentration of the polymer. Therefore it takes a long simulation time to get a huge set of various system configurations. Visual analysis of the trajectories of the systems in vacuum shows that after approximately 5 ns there are no significant changes in the systems configurations. It allows us to conclude that 10 ns is enough time for such types of systems.

  1. In general, the Results and Discussion section could be more easily read if divided into sub-sections.

Author response #6:

We thank Reviewer for the recommendation but it’s rather difficult to divide this text into subsections without a radical revision of the manuscript structure.

  1. In Figure 4, please use the same y-axis range and align sub-figures (a) and (b).

Author response #7:

Figure 4a and 4b were corrected.

  1. In Figure 5(a), it is unclear if there is a red line (System 5) trace at 0, or if there is a reason for the trace to disappear completely.

Author response #8:

In Figure 5(a) in the time interval from 0-0.7 ns, the number of closest contacts equals 0 and the red line coincides with the abscissa and therefore is not visible.

  1. In Figure 5(b), what happens in System 6 at ~7.5 nm that results in a jump in N(C) after a period of stability?

Author response #9:

Visual analysis of trajectory shows that at ~7.5 ns the one else of the three PAM macromolecules undergoes conformational changes and adsorbs on CNC surface. For better understanding, we have added two instant snapshots of System 6 at 7 ns and after the jump in NC(t) at 7.8 ns in Supplementary Material.

Reviewer 2 Report

Review

on manuscript NANOMATERIALS-829768,

entitled “Molecular Dynamics simulation of polyacrylamide adsorption on cellulose materials”.

Overall opinion

The manuscript presents results of Molecular Dynamics simulations studying the adsorption of PAM on cellulose nanocrystals. This is a, computationally challenging problem and this paper addresses a fair part of it. It is within the scope of the Nanomaterials journal and clearly interesting for its audience.

The paper can be considered for publication, if an extended revision takes place.

Detailed analysis of the manuscript – Abstract

The abstract reflects the content of the manuscript pretty well.

Detailed analysis of the manuscript – 1. Introduction

The introduction of the manuscript is short and concise. It would be nice if a more extensive overview of the current (scientific) literature could be offered to the readers, i.e., previous studies, experimental observations and measurements. I would suggest that the authors provide some references to reviews on polymer nanocomposites, in the relevant paragraph for the interested reader to follow (e.g., Macromolecules 2006, 39, 5194; Macromolecules 2007, 40, 8501; J. Polym. Sci. Part B 2008, 46, 2666; Annu Rev Chem Biomol Eng 2010, 1, 37; Arch Computat Methods Eng 2018, 25, 591). These review papers cover many issues on polymer nanocomposites, including previous studies on adsorption to solid surfaces. Moreover, there are some recent works that computationally address part of the problem and they can be cited: Macromolecules 2014, 47, 6964; Macromolecules 2017, 50, 8827; Macromolecules 2019, 52, 7503.

The authors should explicitly point out the similarities/differences of their present work (with respect to the existing literature) and justification of its novelty.

Detailed analysis of the manuscript – 2. Computational Details

As far as the generation of initial configurations is concerned, the authors provide no explanation on how the volume/size/shape of the simulation box is adjusted. Since they do not perform NpT simulations, it is really interesting to explain on how they prepare the initial configurations, given the fact that interfacial systems should be created under the constraint of chemical potential equality.

The authors do not discuss about the equilibration of their systems. However, they do not provide any clear observable proving that the systems have been equilibrated thoroughly. They should include evidence that the structures obtained are equilibrated at all length- and time-scales.

Some minor comments:

- line 51, page 2: "NVT" and all occurrences of variables should appear in italic font, i.e., "NVT".

Detailed analysis of the manuscript – 3. Results and Discussion

Results seem to have been obtained from a single initial configuration for every system. This is unacceptable. All observables of molecular simulations are obtained as ensemble averages over different initial configurations. The authors should comment on whether they have used an ensemble of simulation trajectories or a single one.

Figure 4 presents several pair distribution functions between pairs of PAM atoms and oxygen and hydrogen atoms of water molecules. In subfigure (a) the pair distribution functions do not converge to unity for long distances (if 1 nm can be considered a large distance in systems of this kind). It would be nice if the authors could explain this behavior, or extend the sampling radius for the pair distribution function calculation.

Author Response

Reviewer 2

Overall opinion

The manuscript presents results of Molecular Dynamics simulations studying the adsorption of PAM on cellulose nanocrystals. This is a computationally challenging problem and this paper addresses a fair part of it. It is within the scope of the Nanomaterials journal and clearly interesting for its audience.

The paper can be considered for publication, if an extended revision takes place.

Detailed analysis of the manuscript – Abstract

The abstract reflects the content of the manuscript pretty well.

 Detailed analysis of the manuscript – 1. Introduction

The introduction of the manuscript is short and concise. It would be nice if a more extensive overview of the current (scientific) literature could be offered to the readers, i.e., previous studies, experimental observations and measurements. I would suggest that the authors provide some references to reviews on polymer nanocomposites, in the relevant paragraph for the interested reader to follow (e.g., Macromolecules 2006, 39, 5194; Macromolecules 2007, 40, 8501; J. Polym. Sci. Part B 2008, 46, 2666; Annu Rev Chem Biomol Eng 2010, 1, 37; Arch Computat Methods Eng 2018, 25, 591). These review papers cover many issues on polymer nanocomposites, including previous studies on adsorption to solid surfaces. Moreover, there are some recent works that computationally address part of the problem and they can be cited: Macromolecules 2014, 47, 6964; Macromolecules 2017, 50, 8827; Macromolecules 2019, 52, 7503.

The authors should explicitly point out the similarities/differences of their present work (with respect to the existing literature) and justification of its novelty.

Author response:

The Introduction of the manuscript has expanded by citing relevant literature on polymer nanocomposites including previous studies and recent works. The current work is a continuation of our previous study which was devoted to the investigation of the microscopic mechanism of PVP adsorption on a cellulose nanocrystal and the role of water in this process. For the first time, classical molecular dynamics simulations of  PAM adsorption on hydrophobic and hydrophilic facets of CNC  have been carried out to interpret the mechanism of the polymer interactions with CNC. The results should help to interpret very interesting experimental data obtained (unpublished results).

Detailed analysis of the manuscript – 2. Computational Details

As far as the generation of initial configurations is concerned, the authors provide no explanation on how the volume/size/shape of the simulation box is adjusted. Since they do not perform NpT simulations, it is really interesting to explain on how they prepare the initial configurations, given the fact that interfacial systems should be created under the constraint of chemical potential equality.

  Author response: We agree with the comment. The Section 2 was corrected by adding more detailed information concerning the generation of initial configurations of the systems. In particular, page 2, lines 85-88.

The systems were solvated by water which was preliminarily equilibrated in NpT-ensemble at 298 K and 0.1 MPa. For water molecules, the SPC/E (Extended Simple Point Charge) model [27] was used.

The production run simulations were performed in NVT ensemble in a cubic box with periodic boundary conditions for 10 ns for all systems in a vacuum and binary system PAM-water, 15 ns for large systems with a high concentration of PAM. For the systems with a low PAM concentration, the simulation time was chosen as 60 ns to obtain sufficient statistical data.

Moreover, in order for the potential reader can better understand the size and shape of the system, as well as the ratio of the sizes of the CNC to the total simulation cell size, we have added an instant snapshot of System 3 with solvent molecules (Figure 1i).

The authors do not discuss about the equilibration of their systems. However, they do not provide any clear observable proving that the systems have been equilibrated thoroughly. They should include evidence that the structures obtained are equilibrated at all length- and time-scales.

Author response: In the Section 2 the information about the systems equilibration is on page 2. One of the evidence that the structures obtained are equilibrated at all length- and time-scales is time dependence of temperature. We include such dependencies for all systems in Supplementary Material and added information in the text. In Supplementary Material one can find the time dependencies of temperature for all systems, which are evidence that the structures obtained are equilibrated well at all length- and time-scales (Figure S1).

Some minor comments:

- line 51, page 2: "NVT" and all occurrences of variables should appear in italic font, i.e., "NVT".

Author response:

We agree with the comment. The text was corrected.

Detailed analysis of the manuscript – 3. Results and Discussion

Results seem to have been obtained from a single initial configuration for every system. This is unacceptable. All observables of molecular simulations are obtained as ensemble averages over different initial configurations. The authors should comment on whether they have used an ensemble of simulation trajectories or a single one.

Author response: We cannot completely agree with the postulate of Reviewer that “all observables of molecular simulations are obtained as ensemble averages over different initial configurations”.  From literature it is known that time averaging has been the most prevalent approach in the context of classical MD simulations [M. F. Shlesinger /Statistical mechanics: Exploring phase space // Nature 405(6783), 135–137 (2000). https://doi.org/10.1038/35012197; Kiarash Gordiz, David J. Singh, and Asegun Henry/ Ensemble averaging vs. time averaging in molecular dynamics simulations of thermal conductivity// Journal of Applied Physics, V. 117, 4; 10.1063/1.4906957]. In time averaging, typically the property of interest is calculated from sufficiently long simulation times until convergence is achieved, which is when the result no longer changes significantly with increased simulation time (phase space sampling). All trajectories were sufficiently long so that enough representative conformations have been sampled. Moreover, to be sure correctness of the data obtained in the current study, we carried out a series of different simulations with various initial parameters (including thermostats, barostats, simulation time, time step, initial configurations, ensemble size and so on). In other words, before carrying out a simulation of ten systems, the big work concerning the selection of parameters was done. It seems to be a common practice not only for computational experiments but and for any physical ones.  

Figure 4 presents several pair distribution functions between pairs of PAM atoms and oxygen and hydrogen atoms of water molecules. In subfigure (a) the pair distribution functions do not converge to unity for long distances (if 1 nm can be considered a large distance in systems of this kind). It would be nice if the authors could explain this behavior, or extend the sampling radius for the pair distribution function calculation.

Author response: 1 nm cannot be considered as a large distance in System 2. We have added graph data in subfigure (a) with extended the sampling radius for the pair distribution function. This insertion shows that RDF converges to unity at 1.5 nm.

Reviewer 3 Report

This seems to be a solid, systematic study.

While overall, the logic and language are very good, 2 sentences are broken and should be fixed:

1) page 4, lines 112-115: The beginning makes no sense, probably the first word "Having" should just be removed.

2) page 5, line 135: The sentence in this line makes no sense, it is entirely broken.

Other minor issue:

You may want to extend the caption of figure 3 slightly, like: 

Examples of the time dependence of the number of PAM-PAM (a) and PAM–water (b) hydrogen bonds (nHB) per PAM monomer unit, used to interpret the mean life time of HBs.

Author Response

Reviewer 3

This seems to be a solid, systematic study.

While overall, the logic and language are very good, 2 sentences are broken and should be fixed:

1) page 4, lines 112-115: The beginning makes no sense, probably the first word "Having" should just be removed.

Author response #1:

We agree with the comment. Text was corrected (in the revised manuscript, page 5, line 134).

2) page 5, line 135: The sentence in this line makes no sense, it is entirely broken.

 Author response #2:

We agree with the comment. Text was corrected (in the revised manuscript, page 6, lines 157-158).

  Other minor issue:

You may want to extend the caption of figure 3 slightly, like: 

Examples of the time dependence of the number of PAM-PAM (a) and PAM–water (b) hydrogen bonds (nHB) per PAM monomer unit, used to interpret the mean life time of HBs.

Author response:

We are grateful the Reviewer for this suggestion but to be precise, the autocorrelation function of the parameter characterizing the HB existence between molecules is used to calculate the average hydrogen bond lifetime, and not directly the dependences nHB(t). Therefore we corrected the caption of Figure 3 as follows: Examples of the time dependence of the number of PAM-PAM (a) and PAM–water (b) hydrogen bonds (nHB) per PAM monomer unit.

Round 2

Reviewer 2 Report

The authors have considerably improved their manuscript. I would highly recommend it now for publication in Nanomaterials.

Author Response

We agree with the reviewer's comment.